# Efficient Dual-Function Catalyst: Palladium–Copper Nanoparticles Immobilized on Co-Cr LDH for Seamless Aerobic Oxidation of Benzyl Alcohol and Nitrobenzene Reduction

**DOI:** 10.3390/nano13131956

**Published:** 2023-06-27

**Authors:** Linah A. Alzarea, Mosaed S. Alhumaimess, Ibrahim Hotan Alsohaimi, Hassan M. A. Hassan, M. R. El-Aassar, Amr A. Essawy, Haitham Kalil

**Affiliations:** 1Department of Chemistry, College of Science, Jouf University, Sakaka 2014, Saudi Arabia; lenahadii@gmail.com (L.A.A.); ehalshaimi@ju.edu.sa (I.H.A.); hmahmed@ju.edu.sa (H.M.A.H.); mrelaassar@ju.edu.sa (M.R.E.-A.); aaessawy@ju.edu.sa (A.A.E.); 2Chemistry Department, Cleveland State University, Cleveland, OH 44115, USA; h.kalil@csuohio.edu

**Keywords:** layered double hydroxides, benzyl alcohol oxidation, sol-immobilization method, nitrobenzene reduction, Pd-Cu NPs

## Abstract

Layered double hydroxides (LDHs) present exciting possibilities across various industries, ranging from catalytic applications to water remediation. By immobilizing nanoparticles, LDHs’ characteristics and functionality can be enhanced, allowing for synergetic interactions that further expand their potential uses. A simple chemical method was developed to produce well-dispersed Pd-Cu NPs on a Co-Cr LDH support using a combination of in situ coprecipitation/hydrothermal and sol-immobilization techniques. The Pd-Cu@Co-Cr LDH catalysts was obtained, showing its catalytic activity in promoting the aerobic oxidation of alcohols and enabling the reduction of nitro-compounds through NaBH_4_ mediation. The physicochemical properties of the prepared catalyst were comprehensively investigated utilizing a range of analytical techniques, comprising FTIR, XRD, XPS, TGA, nitrogen adsorption isotherm, FESEM, and HRTEM-EDX. The findings showed the significance of immobilizing the bimetallic Pd-Cu nanoparticles on the Co-Cr LDH via an exceptional performance in the aerobic oxidation of benzyl alcohol (16% conversion, 99.9% selectivity to benzaldehyde) and the reduction of nitrobenzene (98.2% conversion, rate constant of 0.0921 min^−1^). The improved catalytic efficacy in benzyl alcohol oxidation and nitrobenzene reduction on the Pd-Cu@Co-Cr LDH catalyst is attributed to the uniform distribution and small size of the Pd-Cu NPs as active sites on the Co-Cr LDH surface. The prepared catalyst demonstrated exceptional stability during repeated runs. This study paves the way for multiple opportunities in tailoring, producing, and precisely controlling catalysts for various organic transformation reactions.

## 1. Introduction

Industry often relies on oxidation reactions to convert petroleum-based feedstocks into chemical reagents, which are then utilized as precursor materials for various applications. However, they frequently employ harmful and polluting methods that yield hazardous wastes. Because of safety issues associated with the use of combustible organic solvents and environmental concerns regarding the generation of harmful side products, the industrial sector has begun to limit the use of oxidation processes [1]. The concepts of environmentally friendly, sustainability, and chemistry should be rigorously taken into consideration when developing a new oxidation process [2]. The research community has consistently aimed to discover an efficient oxidant that aligns with the aforementioned concepts, as traditional oxidation methods are no longer suitable [3]. Among the most extensively studied processes in synthetic organic chemistry is the oxidation of alcohols to their corresponding acids, ketones, and aldehydes. This is partly because of the widespread presence of these components in fragrances, vitamins, and medications [4].

To accomplish the foregoing efforts, tremendous work has been put into developing aerobic oxidation processes. The most effective substitute for the current approaches is the integration of metallic or bimetallic catalyst and molecular oxygen [5,6]. In fact, oxygen is an economical and low-polluted oxidizing agent with the highest oxygen content. This enables reactions to occur under moderate conditions, thereby conserving the overall energy required [7]. A desirable green approach for the manufacture of the finest chemicals in the industrial field has been identified as the use of molecular oxygen in the integration with heterogeneous catalytic systems. The basis of many current industrial operations is the use of stoichiometric inorganic oxidants, such as K_2_Cr_2_O_7_. Nonetheless, these procedures often generate significant waste and typically exhibit low atomic efficiency, making them ultimately uneconomical and environmentally unfriendly [8,9,10]. Because of the successful implementation of numerous homogeneous catalysts capable of catalyzing liquid phase oxidation, substantial product quantities have been achieved [11,12,13,14]. However, it is well known that isolation and recycling are the main drawbacks of using homogenous catalysis. Thus, research has switched to the design and tailor of heterogeneous catalysts that could be utilized for fixed-bed and batch reactors. The development of numerous catalysts based on multicomponent oxide support has occurred over the past ten years [15,16]. In reality, noble metals serve as the basis for some of the most intriguing heterogeneous catalysts in the field today [17,18]. One drawback of using noble metal catalysts is their tendency to exhibit low selectivity and suffer from catalyst deactivation. However, by utilizing a bimetallic catalyst that incorporates a secondary metal component, it is possible to improve the catalytic activity, selectivity, and durability while reducing the likelihood of deactivation [18]. Interestingly, recent research has demonstrated that bimetallic catalysts are highly effective in facilitating organic transformation reactions [19,20,21,22,23,24,25,26,27,28,29,30,31].

Cobalt’s redox aptitude, which impacts the stability of an oxidation state, is influenced by several factors. These include the shape, particle size, and type of catalytic support used [30,31,32]. Cobalt’s propensity for redox reactions can be altered by incorporating another oxide or metal, commonly referred to as a doping or promoter constituent.

Because of their exceptional physicochemical properties and distinctive 2D structure, layered double hydroxides (LDHs) have recently gained more attention in catalysis [33]. LDHs are materials characterized by a layered structure including positively charged metal cations, negatively charged hydroxide ions, and interlayer anions. These metal cations typically include divalent (II) and trivalent (III) metals, such as magnesium (Mg), aluminum (Al), or zinc (Zn). Within the layered structure, each layer contains a combination of (II) and (III) cations interconnected by hydroxide ions. Positioned between the layers, the interlayer anions can be replaced with other anions, providing unique versatility to the LDH materials [34]. The active sites in LDHs are typically located on the edges or basal planes of the sheets, where the metal cations are exposed and accessible to reactants. The dispersion of active sites refers to the distribution and accessibility of these sites within the LDH structure. The higher the dispersion, the more active sites are available for catalytic reactions, leading to increased catalytic activity. LDHs have been shown to be effective catalysts for a wide range of catalytic reactions, comprising oxidation, reduction, acid–base catalysis, and photocatalysis. The catalytic performance of LDHs can be affected by factors such as the size and composition of the metal cations, the interlayer anions, and the dispersion of active sites [35].

The novelty of the current work lies in the development of a chemical method to create well-dispersed Pd-Cu nanoparticles on a Co-Cr LDH support. This approach combines in situ coprecipitation/hydrothermal and sol-immobilization techniques, resulting in the formation of a Pd-Cu@Co-Cr LDH catalyst. The catalyst demonstrated enhanced catalytic activity in the aerobic oxidation of alcohols and the reduction of nitro compounds. Additionally, the comprehensive investigation of the catalyst’s physicochemical properties using various analytical techniques provides valuable insight into its performance. The improved catalytic efficacy, attributed to the uniform distribution and small size of the PdCu nanoparticles on the Co-Cr LDH surface, along with the exceptional stability during repeated runs, represents the novelty of this study. Additionally, the Co-Cr-LDH catalyst exhibits greater stability to the final matrix because of the resistance of both metals to leaching or deactivation during catalytic reactions. Moreover, the distinct arrangement of Co and Cr ions within the LDH structure influences the availability and distribution of the active sites on the material’s surface. This specific arrangement impacts the dispersion and accessibility of active sites in Co-Cr-LDH, which subsequently affects its catalytic activity and selectivity [36]. These findings pave the way for opportunities in tailoring, producing, and controlling catalysts for diverse organic transformation reactions.

## 2. Materials and Methods

### 2.1. Chemicals

Cobalt (II) nitrate hexahydrate (98%), chromium (III) nitrate (99%), sodium hydroxide (≥98%), copper (II) nitrate (≥99.9%), palladium (II) nitrate solution (99.9%), polyvinyl alcohol (PVA) (10,000, 80%), benzyl alcohol (≥99%), benzyl benzoate (≥99%), benzene (99.8%), benzaldehyde (≥99%), toluene (≥99.7%), benzoic acid (≥99.9%), and nitrobenzene (≥99%) were obtained from Sigma-Aldrich, St. Louis, CO, USA. Sodium borohydride and acetone were bought from Fisher Scientific, Hanover Park, IL, USA.

### 2.2. Materials Fabrication

#### 2.2.1. Synthesis of Pristine Co-Cr LDH

Co(II)-Cr(III) with a molar ratio of 3:1 was adopted to synthesize the bare Co-Cr LDH utilizing the coprecipitation approach under a constant pH value. In brief, under vigorous stirring at ambient temperature, 0.0125 mol Cr(NO_3_)_3_ 9H_2_O and 0.037 mol Co(NO_3_)_2_ 6H_2_O were dissolved in 100 mL DI water. After, NaOH (2M) solution was added dropwise until the pH was brought down to approximately 9.5. The resultant slurry was then treated hydrothermally for approximately 24 h at 80 °C. A centrifuge was used to filter the precipitate, and then it was rinsed with DI water until the pH became neutral. The Co-Cr LDH was finally obtained by drying the precipitate at 80 °C for 18 h [37,38] (Figure 1).

#### 2.2.2. Sol-Immobilization of Pd-Cu NPs over Co-Cr LDH

The detailed catalyst synthesis approach is outlined in the literature [39]. In brief, an aqueous solution of Pd(NO_3_)_2_ (6 mg/mL) and Cu(NO_3_)_2_ 2.5H_2_O (6 mg/mL) was utilized to synthesis 1 wt.% of Cu:Pd catalyst. After stirring the solution of palladium and copper for 20 min, PVA (1 wt.%) was added to the mixture. Afterward, 0.2 M NaBH_4_ was poured into the mixture, leading to a dark brown color that was agitated for 30 min. Then, Co-Cr LDH support was added to the blend and stirred for an additional 60 min. The Pd-Cu@Co-Cr LDH catalyst was obtained through separation, rinsing with deionized water and drying for 12 h at 120 °C. The same procedure was applied to prepare Pd@Co-Cr LDH and Cu@Co-Cr LDH catalysts (Figure 1).

### 2.3. Characterizations

To explore the crystal structure of the fabricated catalysts, an X-ray diffraction (XRD) investigation was performed utilizing a Shimadzu D/Max2500VB2+/Pc with Cu Ka radiation (*λ* = 1.54056 Å). The mean crystalline size of the LDHs was determined by applying the Debye–Scherrer equation:(1)$$D\;(\mathrm{particle}\;\mathrm{size})=\frac{\alpha \lambda \;}{\beta \;cos\theta }$$
where D is the diameter of the LDH particles, *λ* is the wavelength (1.5418 Å) of the X-ray, *α* is assigned to the Scherrer constant (0.9), *θ* is the Bragg’s angle, and *β* is the full-width of the half maximum (FWHM).

Using Bragg’s equation with the (003) reflection, the basal spacing was determined [21] (2):(2)$$d_{\left(hkl\right)}=\frac{n\lambda }{2\;sin\theta }$$
where *n* is a positive integer, *λ* is the wavelength, and *θ* is the scattering angle. Because the LDHs’ structure is hexagonal with a high aspect ratio [22], the lattice parameters “*a*” and “*c*” were evaluated in *a* = *b* ≠ *c*, *α* = *β* = 90°, and *γ* = 120°, using Equation (3) [23].
(3)$$\frac{1}{d_{\left(hkl\right)}^{2}}=\frac{4}{3}\;\left(\frac{h^{2}+hk+k^{2}}{a^{2}}\right)+\frac{l^{2}}{c^{2}}\;$$

The “*a*” and “*c*” lattice variables were computed utilizing the (110) and (003) reflections, respectively. The fabricated materials underwent various analyses to examine their surface function moieties, surface area, porosity, and morphology. Fourier transform infrared (FTIR) spectroscopy was used with a Shimadzu IR Tracer-100 Fourier Transform. X-ray photoelectron spectroscopy (XPS) was performed using a K-ALPHA instrument (ThermoFisher Scientific, Hanover Park, IL, USA), employing monochromatic X-ray Al K-alpha, a full-spectrum pass energy of 200 eV, and a narrow spectrum of 50 eV. The specific surface area and porosity of the materials were determined using Brunauer–Emmett–Teller (BET) and the Barrett–Joyner–Halenda (BJH) methods, respectively, with a NOVA 4200e instrument (Quantachrome Instruments, Boynton Beach, FL, USA) at 77 K. Field emission scanning electron microscopy (FESEM) was used to examine the morphological texture of the materials, employing a Zeiss FESEM Ultra 60. Additionally, high-resolution transmission electron microscopy (HRTEM) images were captured using a JEOL-2011 electron microscope operating at 200 kV.

### 2.4. Catalytic Activity

#### 2.4.1. Benzyl Alcohol Oxidation

The Radleys–Carousel reactor was employed for the liquid-phase oxidation of benzyl alcohol. The developed Cu@Co-Cr LDH, Pd@Co-Cr LDH, and Pd-Cu@Co-Cr LDH catalysts (150 mg) were suspended in (2 g) substrate without the need of any solvents. The mixture was agitated for 4 h at 120 °C at 1200 rpm at a regulated oxygen pressure of 1 bar. The pressure was kept constant throughout the reaction utilizing a pressure gauge to stop any variations in the pressure. A sample was removed from the reactor and examined with gas chromatography (GC). The investigation of the products was conducted using gas chromatography (GC) with a GC-Varian star CP-3800 equipped with a CP-wax 52 column and flame ionization detector (FID). The final products were identified through comparison with established standards. An external standard approach was employed to quantify the number of reactants used and products produced.

#### 2.4.2. Nitrobenzene Reduction

The conversion efficacy of the fabricated materials was tested using NaBH_4_-mediated conversion of nitrobenzene (NB). In brief, the reactor tube was filled with deionized water (5 mL), 0.8 mL of 1.8 × 10^−4^ M nitrobenzene, and 0.005 g of NaBH_4_ solubilized in 1 mL of ice water. Then 1 mg of Pd-Cu@Co-Cr LDH catalyst were introduced to this solution and stirred using a vortex. The reduction process was monitored using UV-Vis spectrometer (Agilent Cary 60) measured at different time intervals to assess the development of the reaction [39,40]. The efficiency of the as-prepared catalysts in the catalytic reduction of nitrobenzene was assessed as follows:(4)$$\mathrm{Conversion}\;(\%)=\left(C_{o}-C_{t}\right)/C_{o}\;\times \;100$$
(5)$$\ln\left(\left[C_{t}\right]\right)/{\left(\left[C_{o}\right]\right)}_{}=-\mathrm{kt}$$
where C_t_ and C_o_ are the nitrobenzene concentration at time t and 0, respectively. The rate constant (k) was estimated from the slope of the linear plot of the natural logarithm of (C_t_/C_o_).

## 3. Results and Discussion

### 3.1. Materials Characterization

#### 3.1.1. Elemental Assessment

The percentages of copper and palladium in all of the LDH samples were determined using ICP analysis (7800 ICP-MS, Agilent, Wood Dale, IL, USA), as tabulated in Table 1. The findings imply that the actual amounts of both Pd and Cu in the synthesized catalysts closely correspond to the calculated values. According to our analysis, the measured values exhibit a slight deviation of only 0.01% to 0.03% from the expected values based on the synthesis process.

#### 3.1.2. Spectroscopic Evaluation

ATR-FTIR analysis was used to further verify the LDH structure by discovering cations and anions in the lamellar structure. This made it easier to comprehend of the composition and arrangement of the LDH and the incorporated ions. The FTIR spectra of Co-Cr LDH (black line), Cu@Co-Cr LDH (red line), Pd@Co-Cr LDH (blue line), and Pd-Cu@Co-Cr LDH (green line) catalysts are illustrated in Figure 1A. The hydroxy moiety is confirmed by the characteristic broad band noted in all of the spectra at 3400–3500 cm^−1^, which is indicative of the stretching vibration of inter-lamellar molecular H_2_O and hydroxyl moiety that are found in layers of LDH. The band’s extreme broadness suggests a diverse range of H-bonds. The weak absorption band at 2398 cm^−1^ is typically associated with the stretching vibration of carbon dioxide. This band is often observed in infrared spectroscopy (IR) analysis of LDH materials. The band that appeared at 1641 cm^−1^ is appointed to the bending vibration of intercalated water molecules. The peak of intercalated water molecules vanished after immobilization of the copper and palladium NPs. The distinctive band at 1347 cm^−1^ is indicative of the stretching vibration of NO_3_^−^. Because of the hydrogen bonding and the effect of the “n” value in NO_3_^−^ (H_2_O)_n_, the band at 1347 cm^−1^ was a little bit wider for the Co-Cr LDH. The incorporation of Cu or Pd NPs resulted in a diminishment in the intensity of the NO_3_^−^ stretching peak with a shift to a higher wavenumber that significantly decreased upon loading of the bimetallic (Cu and Pd) NPs. This demonstrated that the anion exchange is indicative for the fastest NO_3_^−^ rotation of anion [41]. It seems that the change in the Cu sample is more significant compared to the Pd sample. This suggests that the incorporation of copper (Cu) nanoparticles had a more pronounced effect on the NO_3_^−^ stretching peak intensity than the incorporation of palladium (pd) nanoparticles. The greater significance of the change in the Pd sample could be attributed to a few factors: (i) Cu nanoparticles may exhibit electronic properties that enhance their interaction with the NO_3_^−^ species more effectively than Pd nanoparticles. This could result in a more significant alteration of the NO_3_^−^ stretching peak. (ii) Cu nanoparticles might have stronger surface interactions with the Co-Cr LDH matrix compared to Pd nanoparticles. These interactions could lead to more substantial modifications in the chemical environment surrounding the NO3- species, resulting in a larger impact on the spectral characteristics. Furthermore, the vibrations modes (O-M-O, M-O-M, and M-O-H) in the LDH structure is primarily responsible for the other characteristic absorption bands below 750 cm^−1^. These findings confirmed the development of LDH [41,42,43].

The XRD diffraction patterns of the developed Co-Cr LDH materials are illustrated in Figure 1B. All samples depict well-defined diffraction lines that might be pointing to the hydrotalcite-like structure (JCPDS#38–0487). That the diffraction patterns correspond to pristine Co-Cr LDH confirms the development of a hydrotalcite-like structure, where the distinctive diffraction lines at 11.24°, 23.2°, 34.15°, 38.22°, and 60.29° could be attributed to the (003), (006), (009), (012), and (110) planes, respectively [44,45]. The XRD pattern of the Co-Cr LDH depicts a distinctive characteristic represented in its retained original hydrotalcite-like structure after incorporation of the Cu and Pd bimetallic NPs, although the diffraction lines become wider and slightly weak. No additional diffraction lines related to the crystalline phases of the Cu and Pd metallic NPs were noted, demonstrating their extensive dispersion over the LDH surface. The (003) basal diffraction spacing was determined to identify whether the Cu and Pd metallic NPs were intercalated into the layered structure or retained on the LDH surface (Table 2). An interlayer spacing of 0.7472 nm was determined by measuring the (003) basal spacing at 11.24°, suggesting the potential occupancy of NO_3_^−^ and H_2_O species within the interlayer space. The incorporation of Cu and Pd metallic nanoparticles resulted in an increased interlayer space of 0.8621 nm. This change can be attributed to the presence of NO_3_^−^ and H_2_O species in the interlayer space, indicating an interaction between these species and the layered material. The introduction of metallic nanoparticles can facilitate interaction with these interlayer species, either through electrostatic forces or chemical means. Consequently, this interaction leads to the rearrangement and slight separation of the layers, ultimately expanding the interlayer spacing.

The mean crystalline size was determined utilizing the aforementioned Scherrer equation. The mean crystallite sizes for the Co-Cr LDH, Cu@Co-Cr LDH, Pd@Co-Cr LDH, and Pd-Cu@Co-Cr LDH were estimated to be 13.3, 13.1, 11.5, and 10.4 nm, respectively (Table 2). The LDHs synthesized using a 2 M solution of NaOH exhibited smaller crystallite sizes, which can be attributed to the lower concentration of the base. This observation is supported by the results of Reactions (6) to (8), which demonstrated the formation of a clear and intense blue solution when using 2 M NaOH, suggesting the generation of amphoteric components.
Cr(NO_3_)_3_·9H_2_O + 6NaOH → Na_3_[Cr(OH)_6_] + 3NaNO_3_ + 9H_2_O(6)
Co(NO_3_)_2_·6H_2_O + 4NaOH → Na_2_[Co(OH)_4_] + 2NaNO_3_ + 6H_2_O(7)

The development of Co-Cr LDH might be disclosed by the following chemical Equation (8):3Co(NO_3_)_2_·6H_2_O + Cr(NO_3_)_3_·9H_2_O + 6NaOH → Co_3_Cr[(OH)_6_]^+3^ · 3(NO_3_)^−^ · yH_2_O + 6NaNO_3_(8)

Additionally, when the ionic radius of the cobalt and chromium ionic species are similar, fine crystalline LDH can develop using a low concentration of NaOH. In this work, we discovered that Co-Cr LDH also crystallizes more effectively when there are two ions with an identical ionic radius of 0.79 Å for Co(II) and 0.76 Å for Cr(III). The response process described above illustrates one potential cause. The (003) basal spacing and average particle size of the CoCr-LDH structure decreased when 2M NaOH was used in the experiment [46,47,48]. A presumable reason is disclosed in the aforementioned mechanism [49,50,51,52].

To explore the oxidation state of the pristine Co-Cr LDH and Pd-Cu@Co-Cr LDH, XPS experiments were conducted (Figure 2). The survey XPS spectrum and the core-level spectrum (Figure 2A) disclose that the Co-Cr LDH comprised the elements Cr, Co, and O. In the case of the Pd-Cu@Co-Cr LDH (Figure 2A), in the survey spectrum typical distinctive peaks of Pd, Cu, Cr, Co, and O elements can be identified. The Co2p deconvolution spectra of Co-Cr LDH and Pd-Cu@Co-Cr LDH might be modeled by two wider spin-orbit doublets (Figure 2B). The Co-Cr LDH spectrum exhibits two distinct signals at 780.57 and 796.18 eV, corresponding to the Co2p3/2 and Co2p1/2 orbitals, respectively. Additionally, there are minor peaks at 785.87 and 803.23 eV, which are associated with the shoulders of the Co2p3/2 and Co2p1/2 signals. In comparison, the peaks for the Co2p3/2 and Co2p1/2 orbitals in the Pd-Cu@Co-Cr LDH show a shift to 781.31 and 797.54 eV, respectively. This shift indicates a strong interaction between the LDH and the Cu and Pd nanoparticles [49,50,51,52]. The altered positions of these peaks suggest changes in the electronic environment and bonding characteristics of cobalt in the presence of the Pd and Cu nanoparticles. For the Cr2p spectra (Figure 2C), the peak positions of Cr2p3/2 (577.31 eV) and Cr p1/2 (587.41 eV) for the Pd-Cu@Co-Cr LDH also positionally shifted compared with those of the Co-Cr LDH (577.11 and 586.84 eV, respectively). The shift in the peak positions could also be attributed to the robust interaction of the loaded NPs with the LDH. The binding energy of Cr 2p at 577.31 eV could be related to the Cr-O and Cr-OH, which suggest that the majority of the Cr^3+^ (70%) and Cr^6+^ (the remaining 30%) were present. Figure 2D illustrates the division of the O 1s signals into distinct peaks for both Co-Cr LDH and Pd-Cu@Co-Cr LDH. The binding energies of 531.19 eV and 531.69 eV can be attributed to the hydroxyl form (OH-group) for the pure Co-Cr LDH and Pd-Cu@Co-Cr LDH, respectively [53]. The observed peak shift in the Pd-Cu@Co-Cr LDH can be ascribed to the electron transfer occurring between the nanoparticles (NPs) and the LDH matrix. The shifts in the binding peaks of Cr 2p, Co 2p, and O 1s between the pristine LDH and Pd-Cu@Co-Cr LDH indicate the presence of chemical interactions rather than a simple physical mixing state. These position changes suggest alterations in the electronic environment and bonding characteristics within the LDH structure upon the introduction of the Pd and Cu nanoparticles. In the case of the high-resolution XPS spectra of copper (Figure 2E), there were two sets of 2p bands: Cu 2_p3/2_ and Cu 2_p1/2_ at binding energies of 933.9 and 951.7 eV, respectively. This could be attributed to the presence of copper with low valance, which comprises Cu and Cu^+^. Meanwhile, the feeble bands of Cu 2_p3/2_ at 936.1 eV and Cu 2_p1/2_ at 955.1 eV, together with the satellite band at 942.8 eV, are typical features of Cu^2+^, which are the result of the eventual oxidation of Cu/Cu^+^ species in these nanocomposites. In Figure 2F of the Pd3d spectra, the prepared Pd-Cu@Co-Cr LDH displays two sets of peaks for Pd3_d5/2_ and Pd3_d3/2_. The low-energy Pd3_d5/2_ peak deconvoluted into doublets at 335.3 and 336.2 eV, which are associated with a mixture of Pd^0^ and Pd^2+^. On the other hand, the Pd 3_d3/2_ shows two bands at 340.5 and 341.2 eV, which are linked to a mixture of Pd° and Pd^2+^. The atomic surface content is estimated to be 80.4% for Pd^0^ and 17.8% for Pd (II). The presence of PdO_2_ species is likely due to the oxidation of the exposed Pd nanoparticles during preparation, leading to substantial oxidation of the catalyst surface and resulting in the oxide state of Pd, namely, PdO_2_ (1.8%) [40].

#### 3.1.3. TGA Analysis

Thermal stability analysis using TGA (thermogravimetric analysis) was performed on the Co-Cr LDH (black line) and Pd-Cu@Co-Cr LDH (green line) in a temperature range of 25 °C to 600 °C, as shown in Figure 3. The TGA thermogram of the pristine Co-Cr LDH exhibited three-step weight loss. The initial weight loss of approximately 13.4% at approximately 213 °C was attributed to the removal of intercalated and weakly bound physically adsorbed water (H_2_O). As the temperature increased from 213 to 315 °C, a significant weight loss of about 15.4% occurred, which was associated with the loss of nitrate anions. The third weight loss step, observed above 315 °C, was attributed to the dehydroxylation process of the LDH. Similarly, for the Pd-Cu@Co-Cr LDH, the TGA profile also showed three distinct stages of weight loss. The first weight loss, at approximately 213 °C, was attributed to the detachment of loosely bound H_2_O molecules from LDH. The second step, accounting for a weight loss of 15.8%, occurred up to 350 °C and corresponded to the removal of the interlayer of H_2_O and nitrate anions. The final weight loss step (17.29%) up to 600 °C was related to the dehydroxylation process, leading to the destruction of the layered structure. Notably, the TGA curve of the Pd-Cu@Co-Cr LDH exhibited a similar decomposition pattern but with a higher total weight loss (46.59%). This increased weight loss in the TGA curve clearly indicates the formation of the Pd-Cu@Co-Cr LDH nanocomposite.

#### 3.1.4. N_2_ Isotherms

The surface area and porosity of all of these Co-Cr LDH materials were explored with the N_2_ adsorption–desorption isotherms at 77 K and porosity measurements. All of the LDHs possessed type IV isotherms with H3 hysteresis loops, which are characteristics of plate-like particles and mesoporous materials (Figure 4A). As presented in Table 3, with the immobilization of the Cu and Pd NPs, the surface area showed an increasing trend from 22.4 to 37.8 m^2^ g^−1^ with a pore volume ranging from 0.1232 to 0.1397 cm^3^g^−1^. The determined surface area of the NPs@Co-Cr LDH samples was noticeably larger than that of the pristine Co-Cr LDH sample (22.4 m^2^ g^−1^). The observed expansion is attributed to the NPs in the NPs@Co-Cr LDH samples, which may have dispersed uniformly on the surface of the LDHs. This dispersion leads to an increased effective surface area by providing additional exposed surfaces where chemical reactions or interactions can take place. The average pore diameter of the Co-Cr LDH, Cu@Co-Cr LDH, Pd@Co-Cr LDH, and Pd-Cu@Co-Cr LDH pattern was 3.2, 4.4, 4.9, and 5.5 nm, respectively, and distributed in the mesoporous range from 2 to 50 nm, according to the BJH pore size distributions (Figure 4B) estimated from the desorption data. Variations in the pore widths of the LDH samples can arise from chemical interactions between the metal nanoparticles and the layered double hydroxides. The introduction of metal species can lead to structural rearrangements within the LDH lattice, potentially affecting the pore size and accessibility. Chemical reactions occurring during synthesis, such as ion exchange or redox processes, may also influence the pore width of the material.

#### 3.1.5. Surface Analyses

The surface morphology of the pristine Co-Cr LDH, Cu@Co-Cr LDH, Pd@Co-Cr LDH, and Pd-Cu@Co-Cr LDH were identified with the FESEM technique. All of the LDH samples demonstrated hexagonal nanoplates, as illustrated in Figure 5A–D. The nanoplate-shaped morphology remained unchanged after immobilization of the Pd and Cu NPs.

TEM analysis was further performed to explore more detailed structural and constituent details. Figure 6A,B display the TEM, HRTEM, and the particle size distribution of the pristine Co-Cr LDH and Pd-Cu@Co-Cr LDHs. The pristine Co-Cr LDH displays irregular hexagonal nanoplatelets. The LDH interlayers are stacked together. The high-resolution transmission electron microscopy (HRTEM) image reveals lattice spacing corresponding to the reflection planes of the LDH structure. Specifically, the lattice fringe was measured to be 0.23 nm, corresponding to the (015) plane of the Co-Cr LDH. Following the immobilization of the Pd-Cu nanoparticles (NPs) on the LDH surface, distinct dark spots representing the Pd-Cu NPs can be observed to be uniformly distributed across the sample. The size distribution of the Pd-Cu NPs, as shown in the histogram (Figure 6B, inset), indicates a predominant population with an average size of 2.6 nm. Another HRTEM image (Figure 6B) was captured, revealing the lattice fringes corresponding to the different crystal planes. A lattice fringe measurement of 0.23 nm is assigned to the (015) plane of the Co-Cr LDH, while a lattice fringe of 0.21 nm corresponds to the (111) plane of the Pd NPs. These HRTEM observations provide valuable insight into the structural characteristics of the system,

The EDX mapping pictures were utilized to further explore the homogenous distribution of the LDH components, and nearly all of the Co, Cr, Pd, Cu and O elements were found, indicating the effective synthesis of the Co-Cr LDH and Pd-Cu@Co-Cr LDHs (Figure 7A,B). The amount of Co and Cr in the LDH samples was 3:1, which is consistent with the first reaction preparation, as verified by the elemental composition using an EDX assessment. Ultimately, it can be stated that the Co-Cr LDH samples were successfully used to create highly distributed Cu and Pd NP-based catalysts.

### 3.2. Catalytic Activity

#### 3.2.1. Benzyl Alcohol Oxidation

The efficiency of the Pd-based catalysts in oxidizing alcohols, a crucial reaction in organic synthesis, was established. To evaluate the catalytic efficacy of the fabricated catalysts (Co-Cr LDH, Cu@Co-Cr LDH, Pd@Co-Cr LDH, and Pd-Cu@Co-Cr LDH), benzyl alcohol oxidation at 120 °C under green conditions was selected. The findings are tabulated in Table 4. In addition to benzaldehyde, the byproducts detected were toluene, benzene, and benzylbenzoate. All of the LDHs disclosed a robust selectivity for benzaldehyde as the target product, varying from 87% to 99.5% within the allowed reaction time. Immobilization of the metallic NPs significantly enhanced the catalytic efficacy, with the Pd-Cu@Co-Cr LDH exhibiting the greatest conversion (16% conversion with a 99.5% selectivity for benzaldehyde).

The synergistic effect between the Pd and Cu NPs, as determined by the major product selectivity, led to a boost in the conversion and aldehyde selectivity through the inhibition of disproportionation reactions. This resulted in a minimum toluene concentration of approximately 0.5% with the Pd-Cu@Co-Cr LDH. The performance of the bimetallic catalysts showed significant differences compared to that of monometallic catalysts. The efficacy on the bimetallic catalysts was higher than that on the Cu@Co-Cr LDH and Pd@Co-Cr LDH. Generally, in catalysis, the oxidation state of the metal species have a significant impact on their reactivity and selectivity. In the case of copper and palladium, these metals can exist in different oxidation states, which can influence their activity in catalytic reactions. Copper can exist in various oxidation states, comprising Cu(0), Cu(I), and Cu(II). Cu(0) is metallic copper with no charge, while Cu(+) is a monovalent copper ion with a positive charge. The oxidation state of copper can influence its activity in catalysis, with Cu(0) often exhibiting superior catalytic activity compared to other oxidation states. Similarly, palladium can also exist in different oxidation states, including Pd(0), Pd(II), and Pd(IV). Pd(0) is metallic palladium with no charge, while Pd(II) is a divalent palladium ion with a positive charge. The oxidation state of palladium can also have a significant impact on its catalytic activity, with Pd(0) often exhibiting superior activity compared to Pd(II) and other oxidation states. In our case, the existence of both Cu(0) and Cu(+) in the catalyst and both Pd(0) and Pd(II) suggests that these metals could be playing different roles in the catalytic reaction. Additionally, the sol immobilization approach produced particles with a smaller size (approximately 2.6 nm) and a narrower size distribution of the particle, which was anticipated to affect the catalyst activity. Therefore, we believe that the distribution of the particle size is the main parameter controlling the catalyst efficacy. The results were compared with previous studies and are presented in Table 5. Notably, under comparable reaction conditions, the bimetallic combination of Cu and Pd supported on Co-Cr LDH exhibited the highest efficacy in the alcohol oxidation, surpassing other previously reported catalytic materials [51,52,53,54,55]. One potential reaction pathway for the alcohol oxidation over the surface of bimetallic nanoparticles (Pd-Cu) is as follows: the surface reaction of the O-H species from the alcohol with the unsaturated edge atoms of Pd(0) and Cu(0) produces a bimetallic alcoholate. This alcoholate then undergoes β-hydride removal to produce the aldehyde and hydride components in the subsequent step. Finally, the hydride component reacts with dioxygen to regenerate the zero-valence metallic nanoparticles, resulting in the liberation of H_2_O.

Under green conditions, the catalysts’ reusability was investigated at 1 bar of O_2_ for 4 h at 120 °C. Throughout four consecutive cycles using the Pd-Cu@Co-Cr LDH catalyst, there was a noticeable reduction in the catalytic performance from 16% to 13%, while the benzaldehyde selectivity was close to 99.9% in the 1st, 2nd, 3rd, and 4th cycles (Figure 8a). The cause of the noticeable drop in the catalytic efficacy could be imputed to a little loss of the catalyst during the separation step. We tested the stability processes of the spent catalyst after the 4th consecutive catalytic cycles with XRD, as shown in Figure 8b. As demonstrated in Figure 8b, the XRD analysis revealed that the patterns of the catalyst before and after the fourth consecutive cycles were identical, indicating the high durability of the catalyst. Furthermore, the heterogeneous nature of the benzyl alcohol oxidation at 120 °C using Pd-Cu@Co-Cr LDH was investigated through hot-filtration experiments, as shown in Figure 8c. To determine this, the reaction was conducted under optimal conditions, and after 60 min, the catalyst was filtered out, allowing the reaction to continue in the filtrate under the same conditions. The findings indicate that there was no remarkable increase in the conversion percentage after the removal of the Pd-Cu@Co-Cr LDH catalyst. This outcome confirms that the active sites were not detached from the Pd-Cu@Co-Cr LDH into the solution and that the observed catalysis was genuinely heterogeneous and occurred only on the catalyst’s surface.

#### 3.2.2. Catalytic Activity for Nitrobenzene (NB) Reduction

The catalytic performance of the Co-Cr LDH, Cu@Co-Cr LDH, Pd@Co-Cr LDH, and Pd-Cu@Co-Cr LDH nanocomposites were quantitatively explored with a reduction of the NB into aniline (AN) by NaBH_4_ as a model reaction. The time-dependent UV-Vis absorption spectra might be used to readily monitor this reaction, as depicted in Figure 9. The blend of NaBH_4_ and NB exhibited a prominent band at 270 nm in the absence of the catalysts. The inclusion of the catalysts in the reaction mixture led to a rapid decline in the absorption band of NB at 270 nm, while the band at 230 nm corresponding to the reduced component, aniline, increased. Because the initial concentration of NaBH_4_ was considerably higher than that of nitrobenzene, the catalytic rate constant was determined using a pseudo-first-order model. Since, the band intensity at 270 nm decreased over time, it was possible to measure the rate constant (k) and the half-life time (t^1/2^) by adopting Equation (5). Figure 9a–d depict the UV-Vis absorption spectra of the NB reduced by NaBH_4_; Figure 9a depicts the respective plots of the ln(C/Co) of NB against time in the presence of the Co-Cr LDH, Cu@Co-Cr LDH, Pd@Co-Cr LDH, and Pd-Cu@Co-Cr LDH nanocomposites; and Figure 9b displays the conversion percentage of NB to AN at various reaction times.

From the linear relationships of ln(Ct/Co), depicted in Figure 10a, the greatest rate constant, k, for this reaction was 0.0921 min^−1^ for the Pd-Cu@Co-Cr LDH catalyst, which is extremely high compared to 0.0073 min^−1^ for the pristine Co-Cr LDH. All of the nanocomposites showed strong linear correlations of ln(C_t_/C_o_) against time (R^2^ > 0.955), supporting the pseudo-first-order kinetics (Figure 10a). It should be noted that the Pd-Cu@Co-Cr LDH displayed the highest catalytic performance with only 15 min to attain a 98.2% conversion (Figure 10b), while the Co-Cr LDH exhibited the smallest catalytic performance and was unable to completely reduce NB even with 30 min (Figure 9b). This research shows that the integrated impact of the bimetallic Pd-Cu alloy-based nanocomposite created a unique platform for superior catalytic performance towards the NB reduction. The enhanced catalytic performance resulted from the great adsorption affinity of the LDH for NB, which resulted in a high content of NB close to the Pd-Cu and boosted the rate of reduction. As a result, both the Co-Cr LDH and bimetallic NPs were necessary for the catalytic activity of the prepared nanocomposites. The findings demonstrated that the reduction pathway occurred under the influence of stirring nitrophenol, NaBH_4_, and Pd-Cu@Co-Cr LDH catalyst. This mechanism involves the attachment of the acceptor moiety (nitrophenolate ion) to the catalyst’s surface and the generation of active hydrogen by the donor BH_4_^−^. The produced active hydrogen converts the bound nitrophenolate species to amino derivatives and, ultimately, the resulting 4-aminophenol detaches from the Pd-Cu@Co-Cr surface. The LDH facilitates efficient electron transfer from the boron hydride ion (BH_4_^−^) to the bimetallic surface, while the bimetallic NPs (Pd-Cu) provide the necessary active hydrogen for the reduction of the nitrophenolate ion to 4-AP.

## 4. Conclusions

Herein, we presented a facile and promising approach for synthesizing Co-Cr LDH, Cu@Co-Cr LDH, Pd@Co-Cr LDH, and Pd-Cu@Co-Cr LDH nanocomposites. The catalysts were fabricated using a coprecipitation/hydrothermal strategy, followed by sol-immobilization to deposit the NPs. The resulting nanocomposites exhibited superior catalytic efficacy and durability in benzyl alcohol oxidation and NaBH_4_-mediated nitrobenzene reduction due to the synergistic interaction between the Pd-Cu NPs and LDHs. Furthermore, the catalyst’s exceptional stability during repeated runs emphasizes its practicality and applicability in various industrial processes. Overall, the Pd-Cu@Co-Cr LDH catalyst demonstrated not only strong activity in reducing nitro compounds but also improved functioning in the aerobic oxidation of benzyl alcohol, emphasizing the significance of the LDH nanostructure.

## Data Availability

Not applicable.

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
