# Peer review of "Efficient Dual-Function Catalyst: Palladium–Copper Nanoparticles Immobilized on Co-Cr LDH for Seamless Aerobic Oxidation of Benzyl Alcohol and Nitrobenzene Reduction"

_nanomaterials, 2023, doi:10.3390/nano13131956_

Round 1

Reviewer 1 Report

The manuscript by Alzarea, Linah A. et al reported Co-Cr LDH supported Pd-Cu NPs for Aerobic Oxidation of Benzyl Alcohol and Nitrobenzene Reduction. Their data showed the Co-Cr LDH supported Pd-Cu NPs with an nice catalytic  performance in the aerobic oxidation of benzyl alcohol (16% conversion, 99.9% selectivity to ben-23 zaldehyde) and the reduction of nitrobenzene (98.2% conversion, rate constant of 0.0921 min-1). However, the data quality in this work was lower than expected. Thus, this paper is not recommended for publication in Nanomaterials.

 More specific comments:

1.  In figure 2e, the data quality is in low signal-to-noise ratio. I cannot believe that is a high-resolution Cu 2p XPS data. Thus, the further analysis on this data is useless.

2. In Table 3, why NPs@Co-Cr LDH samples showing a noticeably bigger surface area than that of the pristine Co-Cr LDH sample?

3. Why the N2 sorption isotherms on Cu@Co-Cr LDH (Red curves) look weird, very different from the others?

4.   In figure 7b, why the XRD patterns of the sample before and after the 4th run look the same including the noise?

5.   In figure 7a, where is the error bars from? And why authors only provided the error bars for selectivity?

6.      Overall, the readability of this work is not strong.

need to be polished

Author Response

Dear Editor

We sincerely thank you and the reviewers for your constructive comments on our manuscript.

Ms. Ref. No.:  nanomaterials-2435285

Title: "Efficient Dual-Function Catalyst: Palladium-Copper Nanoparticles
Immobilized on Co-Cr LDH for Seamless Aerobic Oxidation of Benzyl Alcohol and
Nitrobenzene Reduction”

Responding to the comments, we have carefully checked all valuable criticisms and suggestions from the referees and editor, and have made suitable revision. Attached please find our point-to-point answers to the referees and the revised version we made accordingly. For clearness reason, the revision parts were marked with red color which we send as review only materials. Thank you for your kind consideration of this paper.

Looking forward to hearing from you soon,

 Sincerely yours,

Mosaed S. Alhumaimess

Response to Reviewer #1

The manuscript by Alzarea, Linah A. et al reported Co-Cr LDH supported Pd-Cu NPs for Aerobic Oxidation of Benzyl Alcohol and Nitrobenzene Reduction. Their data showed the Co-Cr LDH supported Pd-Cu NPs with an nice catalytic performance in the aerobic oxidation of benzyl alcohol (16% conversion, 99.9% selectivity to ben-23 zaldehyde) and the reduction of nitrobenzene (98.2% conversion, rate constant of 0.0921 min-1). However, the data quality in this work was lower than expected. Thus, this paper is not recommended for publication in Nanomaterials.

Response: We appreciate the reviewer's valuable feedback and thank them for taking the time to provide a careful observation and review of our work. While we agree with the reviewer's suggestion. Initially, our work describes the aerobic oxidation of benzyl alcohol in a free solvent, which is different from other references those uses organic solvents. While the reviewer suggests that our reported conversion and selectivity rates do not necessarily indicate exceptional catalyst performance, we would like to emphasize that our study focused on a specific reaction condition without a solvent. Moreover, our goal was not to compare the performance of different catalysts, but rather to investigate the performance of our synthesized catalyst under our specific reaction conditions. Furthermore, it is important to note that the main focus and objective of this study is to investigate the potential of free solvent oxidation as a method for synthesizing organic compounds, particularly in the oxidation of benzyl alcohol. Therefore, the study aims to demonstrate the advantages of this green chemistry process over traditional methods and to highlight its potential for wider applications in organic synthesis. The free solvent oxidation of benzyl alcohol has several advantages over traditional methods. Firstly, it is a green chemistry process that eliminates the need for toxic or hazardous solvents, making it more sustainable and environmentally friendly. Secondly, it is a cost-effective process as it does not require expensive solvents or catalysts. Additionally, the mild reaction conditions of the process reduce the risk of unwanted side reactions, resulting in greater selectivity and yield. It is worth mentioning that this study represents a preliminary investigation into the catalytic activity of Co-Cr LDH and Pd-Cu@Co-Cr LDH in the free solvent oxidation of benzyl alcohol. As such, we are continuing to work on optimizing the catalytic conversion of the system through further experimentation in the future work.

More specific comments:

Remark (1): In figure 2e, the data quality is in low signal-to-noise ratio. I cannot believe that is a high-resolution Cu 2p XPS data. Thus, the further analysis on this data is useless.

Response: We apologize for the low signal-to-noise ratio observed in Figure 2e, which affected the data quality. To address this issue, we inadvertently applied smoothing to the figure, which may have inadvertently affected the integrity of the data. In the revised version, we have ensured that the noise is appropriately represented without any additional smoothing. Thank you for bringing this to our attention, and we appreciate your understanding.

Remark (2): In Table 3, why NPs@Co-Cr LDH samples showing a noticeably bigger surface area than that of the pristine Co-Cr LDH sample?

Response: The increased surface area observed in NPs@Co-Cr LDH samples compared to the pristine Co-Cr LDH sample could be attributed to the presence of NPs (nanoparticles) on the surface. LDH stands for layered double hydroxides, which are materials with a layered structure. When NPs are incorporated into the LDH structure, they can create intercalation or exfoliation of the layers, leading to an expanded surface area. Furthermore, the NPs in the NPs@Co-Cr LDH samples may be dispersed uniformly on the surface of the LDH. This dispersion leads to an increased effective surface area by providing additional exposed surfaces where chemical reactions or interactions can take place.

Remark (3): Why the N2 sorption isotherms on Cu@Co-Cr LDH (Red curves) look weird, very different from the others?

Response: In the revised version of the manuscript, we have included the newly obtained N2 sorption isotherms for the Cu@Co-Cr LDH samples. These curves now align more closely with the expected behavior and exhibit consistent trends with the other samples.

Remark (4): In figure 7b, why the XRD patterns of the sample before and after the 4th run look the same including the noise?

Response: I am writing to sincerely apologize for this error in our manuscript. Upon careful review, we discovered that there was an inadvertent duplication of the same data for both figures, depicting the sample before and after the 4th run. We deeply regret this oversight, and I assure you that it was not intentional. We have taken immediate action to rectify this error in the revised version of the manuscript. The correct XRD patterns for the sample before and after the 4th run have been inserted into their respective figures.

Remark (5): In figure 7a, where is the error bars from? And why authors only provided the error bars for selectivity?

Response: The error bars were derived from replicate measurements of the same experiment conducted under identical conditions. Regarding Figure 7a in our manuscript. We apologize for any confusion caused by the absence of error bars in the figure. Upon reviewing the figure, we realized that there was an oversight in not including the error bars in Figure 7a. We sincerely apologize for this omission, as error bars are essential for accurately representing the uncertainty associated with the data. In the revised version of the manuscript, we have added error bars to Figure 7a to provide a comprehensive and precise representation of the data.

Remark (6): Overall, the readability of this work is not strong.

Response: Thank you for your feedback on the readability of our manuscript. We appreciate your valuable input, and we are pleased to inform you that we have thoroughly revised the entire manuscript to enhance its readability and clarity based on your suggestion.

Reviewer 2 Report

In this work, the authors reported the preparation of Pd-Cu@Co-Cr LDH catalyst and its application for the aerobic oxidation of alcohols and enabling the reduction of nitro-compounds through NaBH4 mediation. The study concluded that the synergistic interaction between Pd-Cu NPs and LDH was favorable for the charge transfer and molecule activation to promote the catalytic performance. Although the present topic seems interesting, there is lack of the relevant characterization and discussion on the exploration of fine structure over catalysts and understanding of structure-activity relationships. Therefore, this work does not fulfill the requirement of Nanomaterials unless the following issues can be addressed.

The following are some suggestions that may help enhance the quality of this work.

1. On line 6 of Page 6, the description “Incorporation of Cu or Pd NPs results in a diminish in the intensity of the NO3- stretching peak …… upon loading the bimetallic (Cu and Pd) NPs” is inconsistent with Figure 1A, and why is there a significant difference between the spectrum of Cu@Co-Cr LDH and Pd@Co-Cr LDH? Please check and correct it along with reasonable explanation.

2. On page 6, as discussed in “the interlayer space increased to 0.8621 nm that are attributed to the metallic and bimetallic NPs are intercalated between the LDH layers and immobilized on their surface”, according to the minimal change of interlayer spacing, this conclusion that the metallic and bimetallic NPs are intercalated between the LDH layers is unreasonable, and the size of bimetallic NPs was inconsistent with discussion below based on the analysis results of TEM images. The author should check and revise the discussion.

3. In Figure 2, there is no clear indication of the catalysts corresponding to the a, b spectra, and the shift of the spectra of Co 2p, Cr 2p and O 1s can be clearly observed when comparing the two catalysts. Please supplement corresponding analysis and explanation. And the proportion of different valence states for Cr and species for O has also undergone significant changes. Please provide relevant analysis and discussion, which could be benefit to better exploration and understanding of catalyst structure. At present, there are many works worthy of reference in this respect can be cited, such as SmartMat, 2023, 4, e1117; Acta Physico-Chimica Sinica, 2022, 38, 38, 2005009; Green Energy & Environment, 2022, 10.1016/j.gee.2022.12.003; Transactions of Tianjin University, 2022, 28, 89; CCS Chem., 2023, 5, 851.

4. On Page 8, the discussion of “both materials exhibited similar thermograms with a two-step weight loss of approximately 46.4%” was contradictory with the results presented in the Figure 3, the weight loss ratio of the two in the second stage is not the same. The author should check and revise the discussion.

5. On Page 9, the discussion of “The observed expansion is a result of intercalation of metallic and bimetallic nanoparticles within the interlayer of the LDH” is very unreasonable and suspicious, the author should revise the analysis and discussion.

6. On Page 10, as discussed in “the mean crystal size by FESEM assessment was determined to be about 20–30 nm and 10–23 nm”, however, the layer size of the materials cannot be clearly observed from the Figure 5, the author should provide HRTEM images to obtain the clear characterization results.

7. In Figure 6, the author should provide the measurement spacing of lattice fringes and mark the crystal planes in order to obtain accurate supporting evidence of catalyst structure.

8. On Page 13, as discussed in “the Cu(+) species could be responsible for activating the substrate, while the Pd(0) species could be responsible for catalyzing the reaction itself”, however, the author did not explore the actual reaction process and active site through in situ infrared characterization, thus, the analysis of the mechanism is unconvincing. The author should analyze and discuss the reaction mechanism through relevant characterization.

9. On Page 15, as discussed in “electron movement from LDH to the bimetallic CuPd NPs, enabling the electron uptake by NB molecules”, however, there is no relevant exploration or analysis of electronic interactions in the entire text, please provide relevant analysis and discussion.

10. There are some errors in the entire text, the numbering letter format of the Figures is inconsistent; the “mg/ml” should be “mg/mL” on Page 4; the “LHD” should be “LDH” on Page 5; the “Figure 8” should be “Figure 9” on Page 16.

Needs to be improved. 

Author Response

Dear Editor

We sincerely thank you and the reviewers for your constructive comments on our manuscript.

Ms. Ref. No.:  nanomaterials-2435285

Title: "Efficient Dual-Function Catalyst: Palladium-Copper Nanoparticles
Immobilized on Co-Cr LDH for Seamless Aerobic Oxidation of Benzyl Alcohol and
Nitrobenzene Reduction”

Responding to the comments, we have carefully checked all valuable criticisms and suggestions from the referees and editor, and have made suitable revision. Attached please find our point-to-point answers to the referees and the revised version we made accordingly. For clearness reason, the revision parts were marked with red color which we send as review only materials. Thank you for your kind consideration of this paper.

Looking forward to hearing from you soon,

 Sincerely yours,

Mosaed S. Alhumaimess

Response to Reviewer #2

Remark (1): On line 6 of Page 6, the description “Incorporation of Cu or Pd NPs results in a diminish in the intensity of the NO3- stretching peak …… upon loading the bimetallic (Cu and Pd) NPs” is inconsistent with Figure 1A, and why is there a significant difference between the spectrum of Cu@Co-Cr LDH and Pd@Co-Cr LDH? Please check and correct it along with reasonable explanation.

Response: Incorporation of Cu or Pd NPs results in a diminish in the intensity of the NO3 stretching peak with a shift to a higher wavenumber that significantly decreased upon loading the bimetallic (Cu and Pd) NPs. This demonstrated the anion exchange is indicative for the fastest NO3rotation of anion [1]. Based on the information you provided, it seems that the change in the Cu sample is more significant compared to the Pd sample. This suggests that the incorporation of copper (Cu) nanoparticles has a more pronounced effect on the NO3- stretching peak intensity than the incorporation of palladium (pd) nanoparticles. The greater significance of the change in the Pd sample could be attributed to a few factors: (i) Cu nanoparticles may exhibit electronic properties that enhance their interaction with the NO3- species more effectively than Pd nanoparticles. This could result in a more significant alteration of the NO3- stretching peak. (ii) Cu nanoparticles might have stronger surface interactions with the Co-Cr LDH matrix compared to Pd nanoparticles. These interactions could lead to more substantial modifications in the chemical environment surrounding the NO3- species, resulting in a larger impact on the spectral characteristics.

  1. J. Goebbert, E. Garand, T. Wende, R. Bergmann, G. Meijer, K.R. Asmis, D.M. Neumark, Infrared spectroscopy of the microhydrated nitrate ions NO3-(H2O) 1-6, J. Phys. Chem. A 113 (2009) 7584–7592.

Remark (2): On page 6, as discussed in “the interlayer space increased to 0.8621 nm that are attributed to the metallic and bimetallic NPs are intercalated between the LDH layers and immobilized on their surface”, according to the minimal change of interlayer spacing, this conclusion that the metallic and bimetallic NPs are intercalated between the LDH layers is unreasonable, and the size of bimetallic NPs was inconsistent with discussion below based on the analysis results of TEM images. The author should check and revise the discussion.

Response: Thank you for pointing out the shortcomings in our work and helping us improve the credibility of our characterization results. In response to your feedback, the section was revised as the following” The interlayer spacing of 0.7472 nm was determined by measuring the (003) basal spacing at 11.24°, suggesting the potential occupancy of NO3 and H2O species within the interlayer space. The incorporation of Cu and Pd metallic nanoparticles resulted in an increased interlayer space of 0.8621 nm. This change can be attributed to the presence of NO3 and H2O species in the interlayer space, indicating an interaction between these species and the layered material. The introduction of metallic nanoparticles can facilitate interaction with these interlayer species, either through electrostatic forces or chemical means. Consequently, this interaction leads to rearrangement and slight separation of the layers, ultimately expanding the interlayer spacing”.

Remark (3): In Figure 2, there is no clear indication of the catalysts corresponding to the a, b spectra, and the shift of the spectra of Co 2p, Cr 2p and O 1s can be clearly observed when comparing the two catalysts. Please supplement corresponding analysis and explanation. And the proportion of different valence states for Cr and species for O has also undergone significant changes. Please provide relevant analysis and discussion, which could be benefit to better exploration and understanding of catalyst structure. At present, there are many works worthy of reference in this respect can be cited, such as SmartMat, 2023, 4, e1117; Acta Physico-Chimica Sinica, 2022, 38, 38, 2005009; Green Energy & Environment, 2022, 10.1016/j.gee.2022.12.003; Transactions of Tianjin University, 2022, 28, 89; CCS Chem., 2023, 5, 851.

Response: We appreciate the reviewer's valuable feedback and thank them for taking the time to provide a careful observation and review of our work. We have taken their valuable observations into consideration and have made the necessary revisions to the section discussing the XPS results, as per their suggestion. Additionally, we have included the relevant references [50-53]in the revised version to strengthen the credibility of our findings.

Remark (4): On Page 8, the discussion of “both materials exhibited similar thermograms with a two-step weight loss of approximately 46.4%” was contradictory with the results presented in the Figure 3, the weight loss ratio of the two in the second stage is not the same. The author should check and revise the discussion.

Response: In the revised version, the discussion pertaining to TGA (Thermogravimetric Analysis) was thoroughly reviewed and corrected.

Remark (5): On Page 9, the discussion of “The observed expansion is a result of intercalation of metallic and bimetallic nanoparticles within the interlayer of the LDH” is very unreasonable and suspicious, the author should revise the analysis and discussion.

Response: The statement was checked in the revised version

Remark (6): On Page 10, as discussed in “the mean crystal size by FESEM assessment was determined to be about 20–30 nm and 10–23 nm”, however, the layer size of the materials cannot be clearly observed from the Figure 5, the author should provide HRTEM images to obtain the clear characterization results.

Response: I would like to express my gratitude for careful observation and valuable input. Thanks to your feedback, we have made significant revisions to the discussion of Field-Emission Scanning Electron Microscopy (FESEM) in the revised version. Additionally, we have included High-Resolution Transmission Electron Microscopy (HRTEM) data to further enhance the comprehensiveness of our study.

Remark (7): In Figure 6, the author should provide the measurement spacing of lattice fringes and mark the crystal planes in order to obtain accurate supporting evidence of catalyst structure.

Response: The lattice fringes was provided in the revised version.

Remark (8): On Page 13, as discussed in “the Cu(+) species could be responsible for activating the substrate, while the Pd(0) species could be responsible for catalyzing the reaction itself”, however, the author did not explore the actual reaction process and active site through in situ infrared characterization, thus, the analysis of the mechanism is unconvincing. The author should analyze and discuss the reaction mechanism through relevant characterization.

Response: Based on your suggestion, we have revised the reaction mechanism by incorporating the relevant characterization.

Remark (9): On Page 15, as discussed in “electron movement from LDH to the bimetallic CuPd NPs, enabling the electron uptake by NB molecules”, however, there is no relevant exploration or analysis of electronic interactions in the entire text, please provide relevant analysis and discussion.

Response: We appreciate the reviewer's feedback and agree that the initial statement regarding electron movement from LDH to the bimetallic CuPd NPs lacked relevant exploration and analysis in the original text. The statement was checked in the revised version based on the obtained results.

Remark (10): There are some errors in the entire text, the numbering letter format of the Figures is inconsistent; the “mg/ml” should be “mg/mL” on Page 4; the “LHD” should be “LDH” on Page 5; the “Figure 8” should be “Figure 9” on Page 16.

Response: We appreciate the reviewer's keen observation and valuable feedback. We have carefully addressed the issues raised and made the necessary corrections

Reviewer 3 Report

This work reports a facile and promising approach for synthesizing Co-Cr LDH, Cu@Co-Cr LDH, Pd@Co-Cr LDH, and Pd-Cu@Co-Cr LDH nanocomposites. In addition, the Pd-Cu@Co-Cr LDH catalyst demonstrated strong activity in reducing nitro compounds and improved functioning in the aerobic oxidation of benzyl alcohol. I thought it appropriate for this manuscript to be published at Nanomaterials, but the following issues must be addressed before acceptance.

1.      The novelty of the work should be explained directly and significantly in introduction.

2.      The “catalytic performance of LDHs can be affected by factors such as the size and composition of the metal cations, the interlayer anions, and the dispersion of active sites” mentioned in the article should be supported by references.

3.      The characterizations for TG or TGA should be detailed in 2.3.

4.      In 3.1.2. Spectroscopic evaluation, the truncation at 2000-2500 cm-1 in FT-IR spectra (A) for the four kinds of catalysts should not be canceled.

5.      In Table 3, what is the concrete meaning for the corner mark? And what is the reason for the changing of the pore width in Figure 4?

6.      Figure 7. (b) hot filtration test should be Figure 7. (c).

7.      Do you think this seamless aerobic oxidation of benzyl alcohol and nitrobenzene reduction method has an industrial application?

Author Response

Response to Reviewer #3

Remark (1): The novelty of the work should be explained directly and significantly in introduction.

Response: The revised version of the work highlighted the novelty based on your suggestion.

Remark (2): The “catalytic performance of LDHs can be affected by factors such as the size and composition of the metal cations, the interlayer anions, and the dispersion of active sites” mentioned in the article should be supported by references.

Response: The statement was supported with reference and cites it as [35] in the revised version.

Remark (3): The characterizations for TG or TGA should be detailed in 2.3.

Response: In the revised version, the discussion pertaining to TGA (Thermogravimetric Analysis) was thoroughly reviewed and corrected.

Remark (4): In 3.1.2. Spectroscopic evaluation, the truncation at 2000-2500 cm-1 in FT-IR spectra (A) for the four kinds of catalysts should not be canceled.

Response: In response to your comment, we have revised the manuscript accordingly, and the truncation at 2000-2500 cm-1 in the FT-IR spectra (A) for all four catalysts has been retained. We apologize for any confusion caused by the initial oversight and appreciate your keen observation. Your guidance has significantly improved the accuracy and completeness of our analysis.

Remark (5): In Table 3, what is the concrete meaning for the corner mark? And what is the reason for the changing of the pore width in Figure 4?

Response: The meaning of the corner mark is “Barrett-Joyner-Halenda (BJH)”. Pore width variations in LDH samples can arise from chemical interactions between the metal nanoparticles and the layered double hydroxide. The introduction of metal species can lead to structural rearrangements within the LDH lattice, potentially affecting pore size and accessibility. Chemical reactions occurring during synthesis, such as ion exchange or redox processes, may also influence the pore width of the material.

Remark (6): Figure 7. (b) hot filtration test should be Figure 7. (c).

Response: The figure title was corrected in the revised version.

Remark (7):     Do you think this seamless aerobic oxidation of benzyl alcohol and nitrobenzene reduction method has an industrial application?

Response: In general, the aerobic oxidation of benzyl alcohol typically leads to the formation of benzaldehyde. This process involves the use of an oxidizing agent, such as molecular oxygen (O2), and a catalyst to facilitate the oxidation reaction. The reduction of nitrophenol, on the other hand, typically results in the formation of aminophenol. This reduction can be achieved through various methods, such as catalytic hydrogenation or using reducing agents like sodium borohydride (NaBH4). Both benzaldehyde and aminophenol have various industrial applications. Benzaldehyde is widely used in the production of fragrances, flavors, dyes, and pharmaceuticals, among other chemical compounds. Aminophenol finds applications in the synthesis of dyes, pharmaceuticals, and rubber chemicals, among other uses. Both the aerobic oxidation of benzyl alcohol and the nitrobenzene reduction method generate active intermediates that can find applications in various industrial manufacturing processes. These intermediates hold potential for use in multiple industries due to their reactivity and versatility. By harnessing the reactivity and versatility of these active intermediates, industrial manufacturers can explore innovative pathways for the production of various products, leading to the development of more efficient and sustainable manufacturing processes.

Reviewer 4 Report

This manuscript by Prof. Alhumaimess et al. describes the preparation of layered double hydroxides (LDH) doped with various metallic nanoparticles. In the study, Cu and Pd are used as co dopants. The pristine material and Cu-, Pd- and mixed-Cu/Pd doped version are compared regarding their catalytic performance. From the results, the Cu/Pd-co-doped material displays the best catalytic ability towards aerobic oxidation of benzyl alcohol and nitrobenzene reduction.

The materials are fully characterized. The catalytic performance of the materials are carefully investigated. This Reviewer finds the manuscript is easy and interesting to read. It is recommended for the acceptance of the manuscript to the Journal.

One minor question to the author,

In Fig. 8, the linear relationship of ln(C/Co) of nitrobenzene against time is shown. According to the plot, it is easy to observe a larger slope for the Cu-doped material compared to that for the Cu-Pd co-doped one while the former is reported to have a smaller value. The author shall recheck the math to ensure the correct results.

fine.

Author Response

Response to Reviewer #4

Remark (1):     In Fig. 8, the linear relationship of ln(C/Co) of nitrobenzene against time is shown. According to the plot, it is easy to observe a larger slope for the Cu-doped material compared to that for the Cu-Pd co-doped one while the former is reported to have a smaller value. The author shall recheck the math to ensure the correct results.

 Response: We would like to express our sincere apologies for the mistake in the figure caption in (Figure 8a). We greatly appreciate your feedback and would like to assure you that the issue has been addressed in the revised version.

Round 2

Reviewer 2 Report

I think the authors did a good reply on the comments, I think it can be accepted as it is.